# IMPLEMENTING INDUCTIVE BIAS FOR DIFFERENT NAVIGATION TASKS THROUGH DIVERSE RNN ATTRACTORS

**Tie Xu, Omri Barak**
Rappaport Faculty of Medicine and Network Biology Research Laboratory
Technion, Israel Institute of Technology
Haifa, 320003, Israel
`fexutie@gmail.com, omri.barak@gmail.com`

## ABSTRACT

Navigation is crucial for animal behavior and is assumed to require an internal representation of the external environment, termed a cognitive map. The precise form of this representation is often considered to be a metric representation of space. An internal representation, however, is judged by its contribution to performance on a given task, and may thus vary between different types of navigation tasks. Here we train a recurrent neural network that controls an agent performing several navigation tasks in a simple environment. To focus on internal representations, we split learning into a task-agnostic pre-training stage that modifies internal connectivity and a task-specific Q learning stage that controls the network's output. We show that pre-training shapes the attractor landscape of the networks, leading to either a continuous attractor, discrete attractors or a disordered state. These structures induce bias onto the Q-Learning phase, leading to a performance pattern across the tasks corresponding to metric and topological regularities. By combining two types of networks in a modular structure, we could get better performance for both regularities. Our results show that, in recurrent networks, inductive bias takes the form of attractor landscapes – which can be shaped by pre-training and analyzed using dynamical systems methods. Furthermore, we demonstrate that non-metric representations are useful for navigation tasks, and their combination with metric representation leads to flexibile multiple-task learning.

## 1 INTRODUCTION

Spatial navigation is an important task that requires a correct internal representation of the world, and thus its mechanistic underpinnings have attracted the attention of scientists for a long time (O'Keefe & Nadel, 1978). A standard tool for navigation is a euclidean map, and this naturally leads to the hypothesis that our internal model is such a map. Artificial navigation also relies on SLAM (Simultaneous localization and mapping) which is based on maps (Kanitscheider & Fiete, 2017a). On the other hand, both from an ecological view and from a pure machine learning perspective, navigation is firstly about reward acquisition, while exploiting the statistical regularities of the environment. Different tasks and environments lead to different statistical regularities. Thus it is unclear which internal representations are optimal for reward acquisition. We take a functional approach to this question by training recurrent neural networks for navigation tasks with various types of statistical regularities. Because we are interested in internal representations, we opt for a two-phase learning scheme instead of end-to-end learning. Inspired by the biological phenomena of evolution and development, we first pre-train the networks to emphasize several aspects of their internal representation. Following pre-training, we use Q-learning to modify the network's readout weights for specific tasks while maintaining its internal connectivity.

We evaluate the performance for different networks on a battery of simple navigation tasks with different statistical regularities and show that the internal representations of the networks manifest in differential performance according to the nature of tasks. The link between task performance and network structure is understood by probing networks' dynamics, exposing a low-dimensional

manifold of slow dynamics in phase space, which is clustered into three major categories: continuous attractor, discrete attractors, and unstructured chaotic dynamics. The different network attractors encode different priors, or inductive bias, for specific tasks which corresponds to metric or topology invariances in the tasks. By combining networks with different inductive biases we could build a modular system with improved multiple-task learning.

Overall we offer a paradigm which shows how dynamics of recurrent networks implement different priors for environments. Pre-training, which is agnostic to specific tasks, could lead to dramatic difference in the network's dynamical landscape and affect reinforcement learning of different navigation tasks.

## 2 RELATED WORK

Several recent papers used a functional approach for navigation (Cueva & Wei, 2018; Kanitscheider & Fiete, 2017b; Banino et al., 2018). These works, however, consider the position as the desired output, by assuming that it is the relevant representation for navigation. These works successfully show that the recurrent network agent could solve the neural SLAM problem and that this could result in units of the network exhibiting similar response profiles to those found in neurophysiological experiments (place and grid cells). In our case, the desired behavior was to obtain the reward, and not to report the current position.

Another recent approach did define reward acquisition as the goal, by applying deep RL directly to navigation problems in an end to end manner (Mirowski et al., 2016). The navigation tasks relied on rich visual cues, that allowed evaluation in a state of the art setting. This richness, however, can hinder the greater mechanistic insights that can be obtained from the systematic analysis of toy problems – and accordingly, the focus of these works is on performance.

Our work is also related to recent works in neuroscience that highlight the richness of neural representations for navigation, beyond Euclidian spatial maps (Hardcastle et al., 2017; Wirth et al., 2017).

Our pre-training is similar to unsupervised, followed by supervised training (Erhan et al., 2010). In the past few years, end-to-end learning is a more dominant approach (Graves et al., 2014; Mnih et al., 2013) . We highlight the ability of a pre-training framework to manipulate network dynamics and the resulting internal representations and study their effect as inductive bias.

## 3 RESULTS

### 3.1 TASK DEFINITION

Navigation can be described as taking advantage of spatial regularities of the environment to achieve goals. This view naturally leads to considering a cognitive map as an internal model of the environment, but leaves open the question of precisely which type of map is to be expected. To answer this question, we systematically study both a space of networks – emphasizing different internal models – and a space of tasks – emphasizing different spatial regularities. To allow a systematic approach, we design a toy navigation problem, inspired by the Morris water maze (Morris, 1981). An agent is placed in a random position in a discretized square arena (size 15), and has to locate the reward location (yellow square, Fig 1A), while only receiving input (empty/wall/reward) from the 8 neighboring positions. The reward is placed in one of two possible locations in the room according to an external context signal, and the agent can move in one of the four cardinal directions. At every trial, the agent is placed in a random position in the arena, and the network's internal state is randomly initialized as well. The platform location is constant across trials for each context (see Methods). The agent is controlled by a RNN that receives the proximal sensory input, as well as a feedback of its own chosen action (Fig. 1B). The network's output is a value for each of the 4 possible actions, the maximum of which is chosen to update the agent's position. We use a vanilla RNN (see Appendix for LSTM units) described by:

$$h_{t+1} = \left(1 - \frac{1}{\tau}\right) h_t + \frac{1}{\tau} \tanh\left(W h_t + W_i f(z_t) + W_a A_t + W_c C_t\right) \tag{1}$$

$$Q(h_t) = W_o h_t + b_o \tag{2}$$

where $h_t$ is the activity of neurons in the networks(512 neurons as default), $W$ is connectivity matrix, $\tau$ is a timescale of update. The sensory input $f(z_t)$ is fed through connections matrix $W_s$, and action feedback is fed through $W_a$. The context signal $C_t$ is fed through matrix $W_c$. The network outputs a Q function, which is computed by a linear transformation of its hidden state.

Beyond the basic setting (Fig. 1A), we design several variants of the task to emphasize different statistical regularities (Fig. 1C). In all cases, the agent begins from a random position and has to reach the context-dependent reward location in the shortest time using only proximal input. The "Hole" variant introduces a random placement of obstacles (different numbers and positions) in each trial. The "Bar" variant introduces a horizontal or vertical bar in random positions in each trial. The various "Scale" tasks stretch the arena in the horizontal or vertical direction while maintaining the relative position of the rewards. The "Implicit context" task is similar to the basic setting, but the external context input is eliminated, and instead, the color of the walls indicates the reward position. For all these tasks, the agent needs to find a strategy that tackles the uncertain elements to achieve the goals. Despite the simple setting of the game, the tasks are not trivial due to identical visual inputs in most of the locations and various uncertain elements adding to the task difficulty.

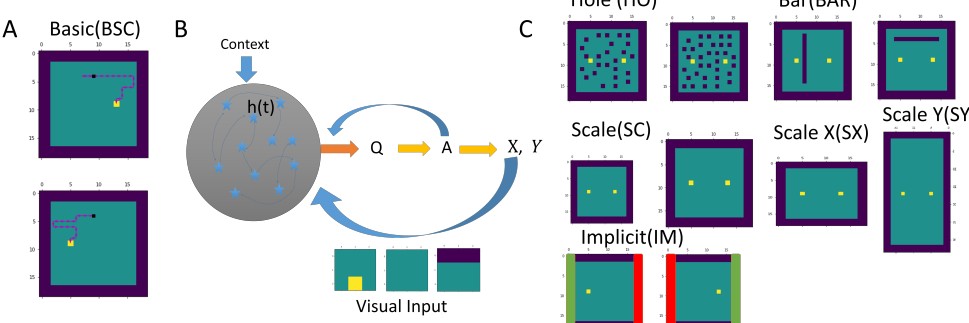

Figure 1: **Navigation task and network architecture(A)** Basic task setting. The agent begins in a random position and has to locate the reward, which is in a context-dependent location. Input is only provided from the 8 neighboring cells. **(B)** The agent is controlled by an RNN that receives input from proximal visual stimuli and its action feedback. An external context is provided to indicate which of two possible reward locations is active. **(C)** Tasks used: Basic, Random obstacles placed in each trial (either holes or bars), scaling the arena in either direction or both, implicit context signal (wall color) instead of external context.

### 3.2 TRAINING FRAMEWORK

We aim to understand the interaction between internal representation and the statistical regularities of the various tasks. In principle, this could be accomplished by end-to-end reinforcement learning of many tasks, using various hyper-parameters to allow different solutions to the same task. We opted for a different approach - both due to computation efficiency (see Appendix III) and due to biological motivations. A biological agent acquires navigation ability during evolution and development, which shapes its elementary cognitive ability such as spatial or object memory. This shaping provides a scaffold upon which the animal could adapt and learn quickly to perform diverse tasks during life. Similarly, we divide learning into two phases, a pre-training phase that is task agnostic and a Q learning phase that is task-specific (Fig. 2A). During pre-training we modify the network's internal and input connectivity, while Q learning only modifies the output.

Pre-training is implemented in an environment similar to the basic task, with an arena size chosen randomly between 10 to 20. The agent's actions are externally determined as a correlated random

walk, instead of being internally generated by the agent. Inspired by neurophysiological findings, we emphasize two different aspects of internal representation - landmark memory (Identity of the last encountered wall) and position encoding (O'Keefe & Nadel, 1978). We thus pre-train the internal connectivity to generate an ensemble of networks with various hyperparameters that control the relative importance of these two aspects, as well as which parts of the connectivity $W, W_a, W_i$ are modified. We term networks emphasizing the two aspects respectively MemNet and PosNet, and call the naive random network RandNet (Fig. 2A). This is done by stochastic gradient descent on the following objective function:

$$S = -\alpha \sum_{i=1}^{n} \hat{P}(z_t) log P(z_t) - \beta \sum_{i=1}^{n} \hat{I}_t log P(I_t) - \gamma \sum_{i=1}^{n} \hat{A}_t log P(A_t) \tag{3}$$

with $z = (x, y)$ for position, $I$ for landmark memory (identity of the last wall encountered), $A$ for action. The term on action serves as a regularizer. The three probability distributions are estimated from hidden states of the RNN, given by:

$$P(I|h_t) = \frac{exp(W_m h_t + b_m)}{\sum_m (exp(W_m h_t + b_m))} \tag{4}$$

$$P(A|h_{t-1}, h_t) = \frac{exp(W_a[h_{t-1}, h_t] + b_a)}{\sum_a exp(W_a[h_{t-1}, h_t] + b_a)} \tag{5}$$

$$P(z|h_t) = \frac{exp((z - (W_p h_t + b_p))^2/\sigma^2)}{\sum_z exp((z - (W_p h_t + b_p))^2/\sigma^2)} \tag{6}$$

where $W_m, W_p, W_a$ are readout matrices from hidden states and $[h_{t-1}, h_t]$ denotes the concatenation of last and current hidden states. Tables 1,2,3 in the Appendix show the hyperparameter choices for all networks. The ratio between $\alpha$ and $\beta$ controls the tradeoff between position and memory. The exact values of the hyperparameters were found through trial and error.

Having obtained this ensemble of networks, we use a Q-learning algorithm with TD-lambda update for the network's outputs, which are Q values. We utilize the fact that only the readout matrix $W_o$ is trained to use a recursive least square method which allows a fast update of weights for different tasks (Sussillo & Abbott, 2009). This choice leads to a much better convergence speed when compared to stochastic gradient descent. The update rule used is:

$$W_o(n+1) = W_o(n) - e(n)P(n)H(n)^T \tag{7}$$
$$P(n+1) = (C(n+1) + \alpha I)^{-1} \tag{8}$$
$$C(n+1) = \lambda C(n) + H(n)^T H(n) \tag{9}$$
$$e(n) = W_o H(n) - Y(n) \tag{10}$$

where $H$ is a matrix of hidden states over 120 time steps, $\alpha I$ is a regularizer and $\lambda$ controls forgetting rate of past data.

We then analyze the test performance of all networks on all tasks (Figure 2B and Table 3 in appendix). Figure 2B,C show that there are correlations between different tasks and between different networks. We quantify this correlation structure by performing principal component analysis of the performance matrix. We find that the first two PCs in task space explain 79% of the variance. The first component corresponds to the difficulty (average performance) of each task, while the coefficients of the second component are informative regarding the nature of the tasks (Fig. 2B, right): Bar (-0.49), Hole(-0.25), Basic(-0.21), Implicit context (-0.12), ScaleX (0.04), ScaleY (0.31), Scale (0.74). We speculate these numbers characterize the importance of two different invariances inherent in the tasks. Negative coefficients correspond to metric invariance. For example, when overcoming dynamic obstacles, the position remains invariant. This type of task was fundamental to establish metric cognitive maps in neuroscience (O'Keefe & Nadel, 1978). Positive coefficients correspond to topological invariance, defined as the relation between landmarks unaffected by the metric information.

Observing the behavior of networks for the extreme tasks of this axis indeed confirms the speculation. Fig. 3A shows that the successful agent overcomes the scaling task by finding a set of actions

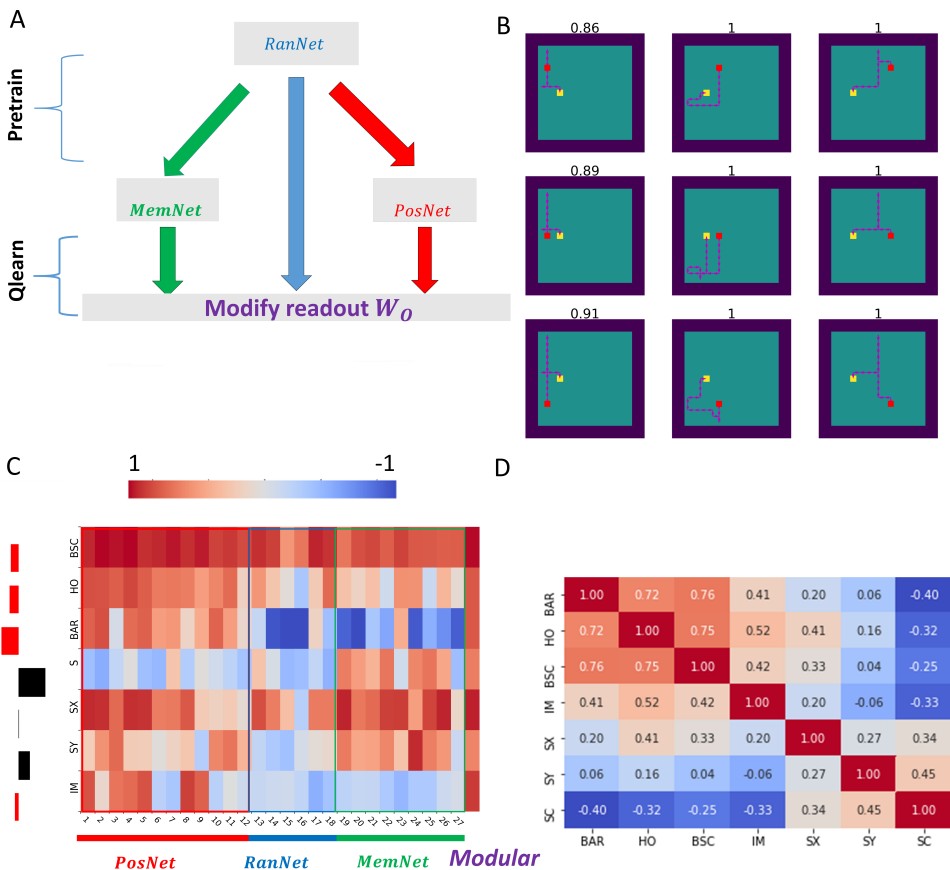

Figure 2: **Training scheme and performance analysis.** (**A**) Two-stage learning framework. Task agnostic pre-training of the internal connectivity is done while emphasizing either position decoding (PosNet) or the identity of the last wall (landmark memory, MemNet). Following pre-training, Q learning of the output is performed for each task. This is also done on networks that were not pre-trained (RandNet). (**B**) Example trajectories of PosNet on the basic task, starting from 9 different initial conditions. The numbers are the scores for each trial (see Appendix). (**C**) Task performance for all networks on all tasks. The score is an average of trials from all starting positions, where each trial is scored by the time relative to the shortest path, or −1 if the agent fails to reach the reward after 120 steps. Bars on the left are coefficients of the second principal component, corresponding to metric vs. topological tasks. The columns show different realizations of Posnet, RanNet and Mem-Net. The last column is a modular network introduced in last section of Results. (**D**) Correlation between all tasks, showing a clustering into two main groups (metric and topological). Parameters for all networks are in Appendix Tables 1,2,3.

that captures the relations between landmarks and reward, thus generalizing to larger size arenas. Fig3B shows that the successful agent in the bar task uses a very different strategy. An agent that captures the metric invariance could adjust trajectories and reach the reward each time when the obstacle is changed. This ability is often related to the ability to use shortcuts (O'Keefe & Nadel, 1978). The other tasks intepolate between the two extremes, due to the presence of both elements in the tasks. For instance, the implicit context task requires the agent to combine landmark memory (color of the wall) with position to locate the reward.

We thus define metric and topological scores by using a weighted average of task performance using negative and positive coefficients respectively. Fig. 3C shows the various networks measured by the two scores. We see that random networks (blue) can achieve reasonable performance with some

hyperparameter choices, but they are balanced with respect to the metric topological score. On the other hand, PostNet networks are pushed to the metric side and MemNet networks to the topological side. This result indicates that the inductive bias achieved via task agnostic pre-training is manifested in the performance of networks on various navigation tasks.

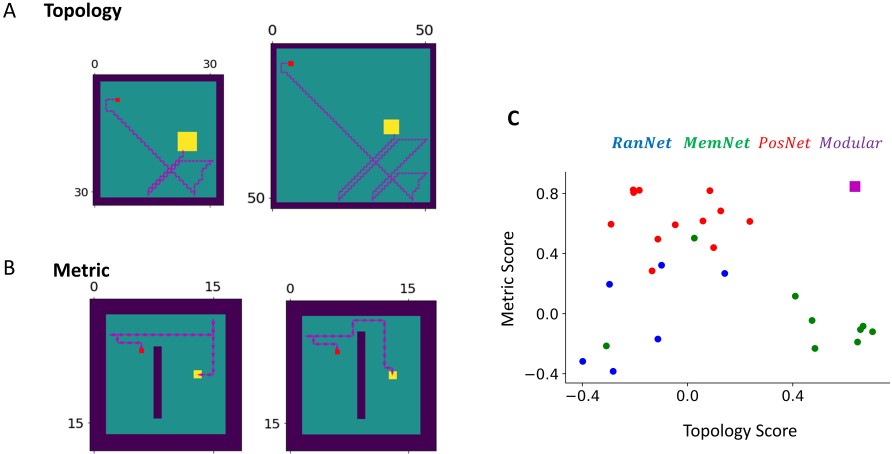

Figure 3: **Different strategies for different regularities.** **(A)** A MemNet network solving the scaling task. The agent uses a sequence of landmark-conditioned actions, and thus generalizes to larger arenas. **(B)** A PosNet network solving the bar task. The agent appears to understand its metric position, and uses it to move in novel paths towards the reward. **(C)** Performance of all networks in all tasks, projected onto the metric and topological scores.

### 3.3 LINKING REPRESENTATION TO DYNAMICS

What are the underlying structures of different networks that encode the bias for different tasks? We approach this question by noting that RNNs are nonlinear dynamical systems. As such, it is informative to detect fixed points and other special areas of phase space to better understand their dynamics. For instance, a network that memorizes a binary value might be expected to contain two discrete fixed points (Fig. 4A). A network that integrates a continuous value might contain a line attractorKim et al. (2017); Kakaria & de Bivort (2017), and , a network that integrates position might contain a plane attractor – a 2D manifold of fixed points – because this would enable updating $x$ and $y$ coordinates with actions, and maintaining the current position in the absence of action (Burak & Fiete, 2009). Trained networks, however, often converge to approximate fixed points (slow points) (Sussillo & Barak; Mante et al., 2013; Maheswaranathan et al., 2019), as they are not required to maintain the same position for an infinite time. We thus expect the relevant slow points to be somewhere between the actual trajectories and true fixed points. We detect these areas of phase space using adaptations of existing techniques (Appendix 5.3, (Sussillo & Barak)). Briefly, we drive the agent to move in the environment, while recording its position and last seen landmark (wall). This results in a collection of hidden states. Then, for each point in this collection, we relax the dynamics towards approximate fixed points. This procedure results in points with different hidden state velocities for the three networks (Fig. 4B) – RandNet does not seem to have any slow points, while PosNet and MemNet do, with MemNet's points being slower. The resulting manifold of slow points for a typical PosNet is depicted in Figure 4C, along with the labels of position and stimulus from which relaxation began. It is apparent that pretraining has created in PosNet a smooth representation of position along this manifold. The MemNet manifold represents landmark memory as 4 distinct fixed points without a spatial representation. Note that despite the dominance of position representation in PosNet, landmark memory still modulates this representation (Fig 3A, M) - showing that pretraining did not result in a perfect plane attractor, but rather in an approximate collection of 4 plane attractors (Fig. 3D, MP). This conjunctive representation can also be appreciated by considering the decoding accuracy of trajectories conditioned on the number of wall encounters. As the agent encounters the wall, the decoding of position from the manifold improves, implying the ability to integrate path integration and landmark memory (Fig Appendix 8).

We thus see that the pre-training biases are implemented by distinct attractor landscapes, from which we could see both qualitative differences between networks and a trade-off between landmark memory and position encoding. The continuous attractors of PosNet correspond to a metric representation of space, albeit modulated by landmark memory. The discrete attractors of MemNet encode the landmark memory in a robust manner, while sacrificing position encoding. The untrained RandNet, on the other hand, has no clear structure, and relies on a short transient memory of the last landmark.

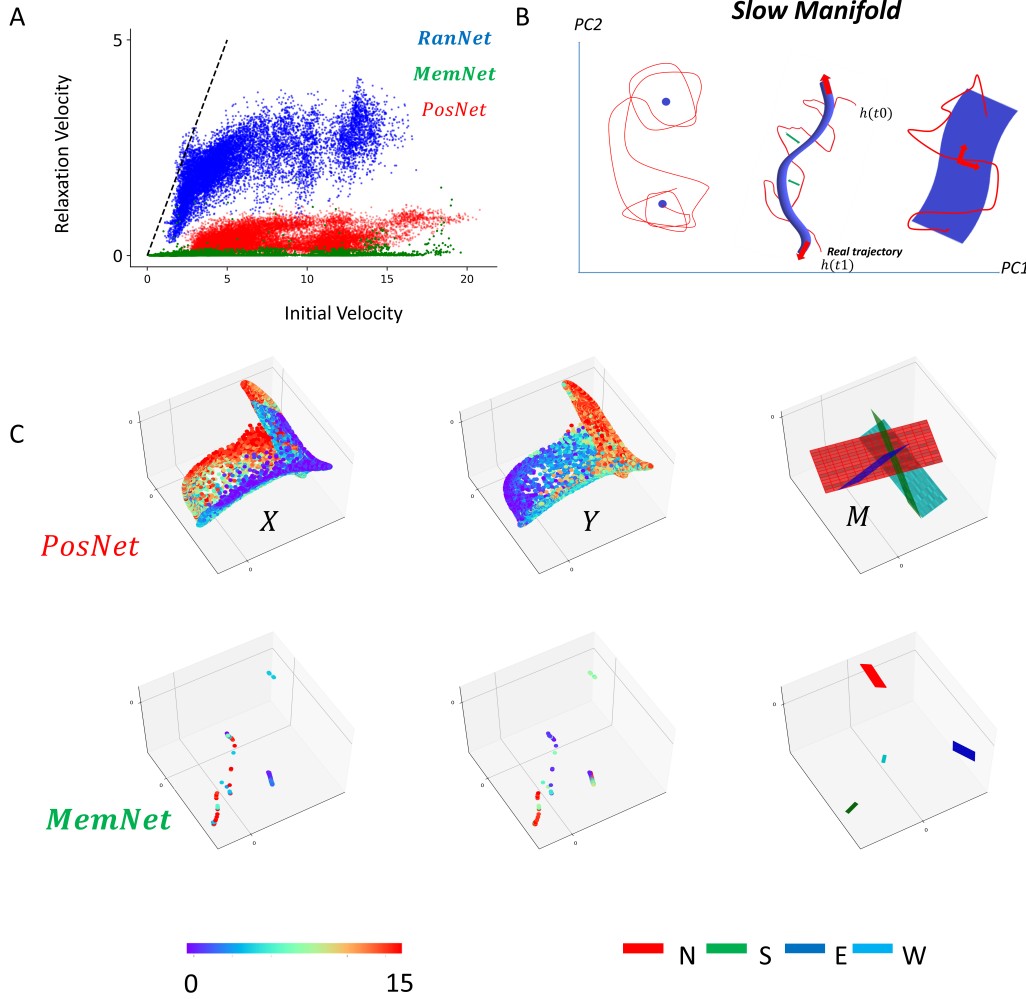

Figure 4: **Diverse attractor landscapes underly diverse agent priors**. **(A)** Velocity of points before and after the relaxation procedure. PosNet and MemNet converged to approximate fixed points (slow points), while RandNet did not. **(B)** Illustration of possible slow point structures (purple), along with trajectories around them (red). A pair of points can encode a discrete memory, a line attractor can integrate a single variable, and a plane attractor can integrate two variables (e.g. $x, y$). **(C)** Attractor landscape for PosNet and MemNet projected into the first 3 PCs of the hidden state. Coloring is according to either X,Y coordinates or the identity of the last wall encountered (landmark memory, M). Note how the position is smoothly encoded on the manifold for PosNet, and memory is encoded by four discrete points for MemNet. The memory panels show a fit of a plane to the X,Y coordinates, conditioned upon a given landmark memory – showing that PosNet also has memory information, and not just position. The networks used are 1,13,20 from Table 3.

The above analysis was performed on three typical networks and is somewhat time-consuming. In order to get a broader view of internal representations in all networks, we use a simple measure of the components of the representation. Specifically, we drove the agent to move in an arena of infinite

size that was empty except a single wall (of a different identity in each trial). We then used GLM (generalized linear model) to determine the variance explained by both position and the identity of the wall encountered from the network's hidden state. Figure 6A shows these two measures for all the networks. The results echo those measured with the battery of 7 tasks (Fig. 3C), but are orders of magnitude faster to compute. Indeed, if we correlate these measures with performance on the different tasks, we see that they correspond to the metric-topological axis as defined by PCA (Fig. 5B, compare with Fig. 2B, right).

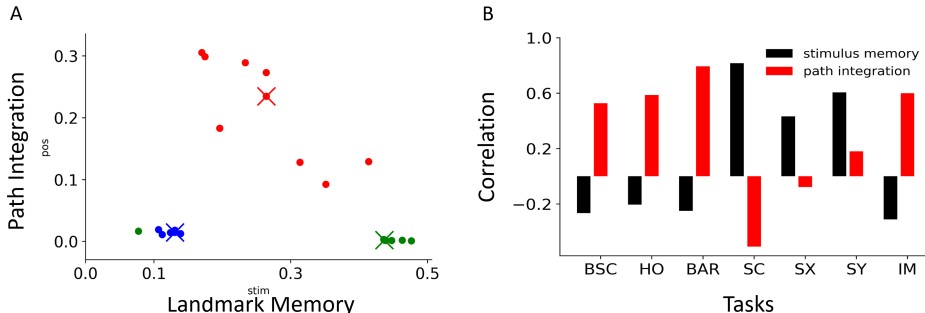

Figure 5: **Components of the internal representation**. **(A)** Variance of the hidden state explained by position and memory for all networks. Note the clear separation between the different pre-training regimes. The crosses denote networks used in Fig. 4. **(B)** Correlation of the two components with performance on all tasks. The strength of the relevant components in the internal representation are predictive of task performance following Q-learning. Note the similarity with the PCA coefficients in Fig. 2B, right.

### 3.4 A MODULAR SYSTEM THAT COMBINES ADVANTAGES OF BOTH DYNAMICS

Altogether, we showed a differential ability of networks to cope with different environmental regularities via inductive bias encoded in their dynamics. Considering the tradeoff is a fundamental property of single module RNN, it is natural to ask if we could combine advantages from both dynamics into a modular system. Inspired by Park et al. (2001). We design a hierarchical system composed of the representation layer on the bottom and selector layer on top. The representation layer concatenates PosNet and MemNet modules together, each evaluating action values according to its own dynamics. The second layer selects the more reliable module based on the combined representation by assigning a value (reliability) to each module. The module with maximal reliability makes the final decision. Thus, the control of the agent shifts between different dynamics according to current input or history. The modular system significantly shifts the metric-topological balance (Fig2C, Fig3B). The reliability V is learned similarly as Q(Appendix 5.7).

$$h_{t+1}^1 = \left(1 - \frac{1}{\tau}\right) h_t^1 + \frac{1}{\tau} \tanh\left(W_{pos} h_t^1 + W_i^1 f(z_t) + W_a^1 A_t + W_c^1 C_t\right) \quad (11)$$

$$h_{t+1}^2 = \left(1 - \frac{1}{\tau}\right) h2_t + \frac{1}{\tau} \tanh\left(W_{mem} h_t^2 + W_i^2 f(z_t) + W_a^2 A_t + W_c^2 C_t\right) \quad (12)$$

$$Q1(h_t^1) = W_o^1 h_t^1 + b_o^1 \quad (13)$$

$$Q2(h_t^1) = W_o^2 h_t^1 + b_o^2 \quad (14)$$

$$V(h_t^1, h_t^2) = W_{sel}([h_t^1, h_t^2]) + b_{sel} \quad (15)$$

## 4 DISCUSSION

Our work explores how internal representations for navigation tasks are implemented by the dynamics of recurrent neural networks. We show that pre-training networks in a task-agnostic manner can shape their dynamics into discrete fixed points or into a low-D manifold of slow points. These

distinct dynamical objects correspond to landmark memory and spatial memory respectively. When performing Q learning for specific tasks, these dynamical objects serve as priors for the network's representations and shift its performance on the various navigation tasks. Here we show that both plane attractors and discrete attractors are useful. It would be interesting to see whether and how other dynamical objects can serve as inductive biases for other domains. In tasks outside of reinforcement learning, for instance, line attractors were shown to underlie network computations (Mante et al., 2013; Maheswaranathan et al., 2019).

An agent that has to perform several navigation tasks will require both types of representations. A single recurrent network, however, has a trade-off between adapting to one type of task or to another. The attractor landscape picture provides a possible dynamical reason for the tradeoff. Position requires a continuous attractor, whereas stimulus memory requires discrete attractors. While it is possible to have four separated plane attractors, it is perhaps easier for learning to converge to one or the other. A different solution to learn multiple tasks is by considering multiple modules, each optimized for a different dynamical regime. We showed that such a modular system is able to learn multiple tasks, in a manner that is more flexible than any single-module network we could train.

Pre-training alters network connectivity. The resulting connectivity is expected to be between random networks (Lukoševičius & Jaeger, 2009) and designed ones (Burak & Fiete, 2009). It is perhaps surprising that even the untrained RandNet can perform some of the navigation tasks using only Q-learning of the readout (with appropriate hyperparameters, see Tables 2,3 and section 4 "Linking dynamics to connectivity" in Appendix). This is consistent with recent work showing that some architectures can perform various tasks without learning (Gaier & Ha, 2019). Studying the connectivity changes due to pre-training may help understand the statistics from which to draw better random networks (Appendix section 4).

Apart from improving the understanding of representation and dynamics, it is interesting to consider the efficiency of our two-stage learning compared to standard approaches. We found that end-to-end training is much slower, cannot learn topological tasks and has weaker transfer between tasks (See Appendix section 5.2). Thus it is interesting to explore whether this approach could be used to accelerate learning in other domains, similar to curriculum learning (Bengio et al., 2009).

### ACKNOWLEDGMENTS

OB is supported by the Israeli Science Foundation (346/16) and by a Rappaport Institute Thematic grant. TX is supported by Key scientific technological innovation research project by Chinese Ministry of Education and Tsinghua University Initiative Scientific Research Program for computational resources.

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

## 5 APPENDIX

### 5.1 PERFORMANCE MEASURE FOR EACH TASK

When testing the agent on a task, we perform a trial for each possible initial position of the agent. Note that the hidden state is randomly initialized in each trial, so this is not an exhaustive search of all possible trial types. We then measure the time it takes the agent to reach the target. This time is normalized by an approximate optimal strategy – moving from the initial position to a corner of the arena (providing x and y information), and then heading straight to the reward. If the agent fails to reach the target after 120 steps, the trial score is $-1$:

$$Score = \left\{ \begin{array}{ll} T/T_{opt} & T < T_{max} \\ -1 & T > T_{max} \end{array} \right. \tag{16}$$

### 5.2 TWO-STAGE VERSUS END-TO-END LEARNING

To check the effectiveness of our two-stage learning (pre-training followed by Q-Learning), we contrast it with an end-to-end approach. We considered two approached to training: a classic deep Q learning version Mnih et al. (2013) and a method adapted from RDPG Heess et al. (2015). The naive Q learning is separated into a play phase and training phase. During the play phase the agent collects experience as $s_t$, $a_t$, $h_t$ $r_t$ into a replay buffer. The action value Q is computed through TD lambda method as $Q(h, a)$ as target function. During the training phase the agent samples the past experience through replay buffer perform gradient descent to minimize the difference between expected $Q(h, a)$ and target Q. We found that for most tasks the deep Q learning method was better than the adapted RDPG, and thus used it for our benchmarks. We used both LSTM and vanilla RNN for these test.

We found that, for the basic task, all networks achieve optimal performance, but our approach is significantly more data-efficient even with random networks (fig. 6A). For all topological tasks, the end-to-end approach fails with both vanilla RNN and LSTM (fig 6C table 3). The end to end approach performs relatively well in metric tasks, except for the implicit context task (fig 6B table 3), which converges to a similar performance as PosNet but with a much slower convergence speed ($10^5$ trials vs $10^4$ trials).

For the end-to-end approaches, a critical question is whether an internal representation emerges, which enables better performance in similar tasks. For instance, do networks that were successfully end-to-end trained in the basic task develop a representation that facilitates learning the bar task? To answer this question, we use networks that were end-to-end trained on one task and use them as a basis for RLS Q-learning of a different task. This allows comparison with the pre-trained networks. Figure 7 shows that pre-training provides a better substrate for subsequent Q-learning - even when considering generalization within metric tasks. For the implicit context task, the difference is even greater.

### 5.3 EXPLORING THE LOW D NETWORK DYNAMICS

Recurrent neural networks are nonlinear dynamical systems. As such, they behave differently in different areas of phase space. It is often informative to locate fixed points of the dynamics, and use their local dynamics as anchors to understand global dynamics. When considering trained RNNs, it is reasonable to expect approximate fixed points rather than exact ones. This is because a fixed point corresponds to maintaining the same hidden state for infinite time, whereas a trained network is only exposed to a finite time. These slow points (Sussillo & Barak; Mante et al., 2013) can be detected in several manners (Sussillo & Barak; Katz & Reggia, 2017). For the case of stable fixed points (attractors), it is also possible to simulate the dynamics until convergence. In our setting, we opt for the latter option. Because the agent never stays in the same place, we relax the dynamics towards attractors by providing as action feedback the average of all 4 actions. The relevant manifold (e.g. a plane attractor) might contain areas that are more stable than others (for instance a few true fixed points), but we want to avoid detecting only these areas. We thus search for the relevant manifold in the following manner. We drive the agent to move in the environment, while recording its position and last seen stimulus (wall). This results in a collection of hidden states, labelled by position

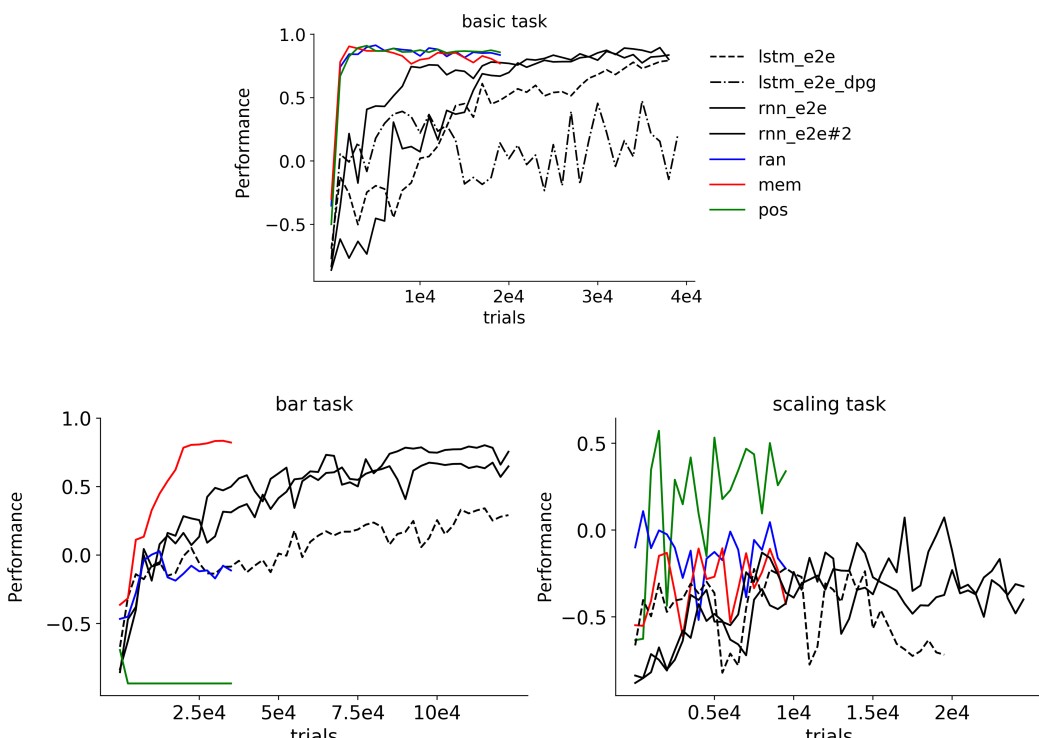

Figure 6: Learning curves for end-to-end training, compared to the Q-learning phase of the two-stage learning. Each curve represents a single network. Curves shown for Basic, Bar and Scaling tasks.

and stimulus that we term the $m = 0$ manifold. For each point on the manifold, we continue simulating the dynamics for $m$ extra steps while providing as input the average of all 4 actions, resulting in further $m \neq 0$ manifolds. If these states are the underlying scaffold for the dynamics, they should encode the position (or memory). We therefore choose $m$ by a cross-validation method – decoding new trajectories obtained in the basic task by using the $k = 15$-nearest neighbors in each $m$-manifold. The red curve in Figure 8A shows the resulting decoding accuracy for position using PosNet, where the accuracy starts to fall around $m = 25$, indicating that further relaxation leads to irrelevant fixed points.

### 5.4 ATTRACTOR LANDSCAPE FOR DIFFERENT RECURRENT ARCHITECTURES

We tested the qualitative shape of the slow manifolds that emerge from pre-training other unit types. Specifically, we pre-trained an LSTM network using the same parameters as PosNet1 and MemNet1 (Table 1). Figures 9 show a qualitatively similar behvior to that described in the main text. Note that MemNet has slow regions instead of discrete points, and we suspect discrete attractors might appear with longer relaxation times. The differences between the columns also demonstrate that slow points, revealed by relaxation, are generally helpful to analyze dynamics of different types of recurrent networks.

### 5.5 PRE-TRAINING PROTOCOLS AND PERFORMANCE OF NETWORKS

As explained in the main text, pre-training emphasizes decoding of either landmark memory or the position of the agent. We used several variants of hyperparameters to pre-train the networks. Equation 17, which is written again for convenience, defines the relevant parameters:

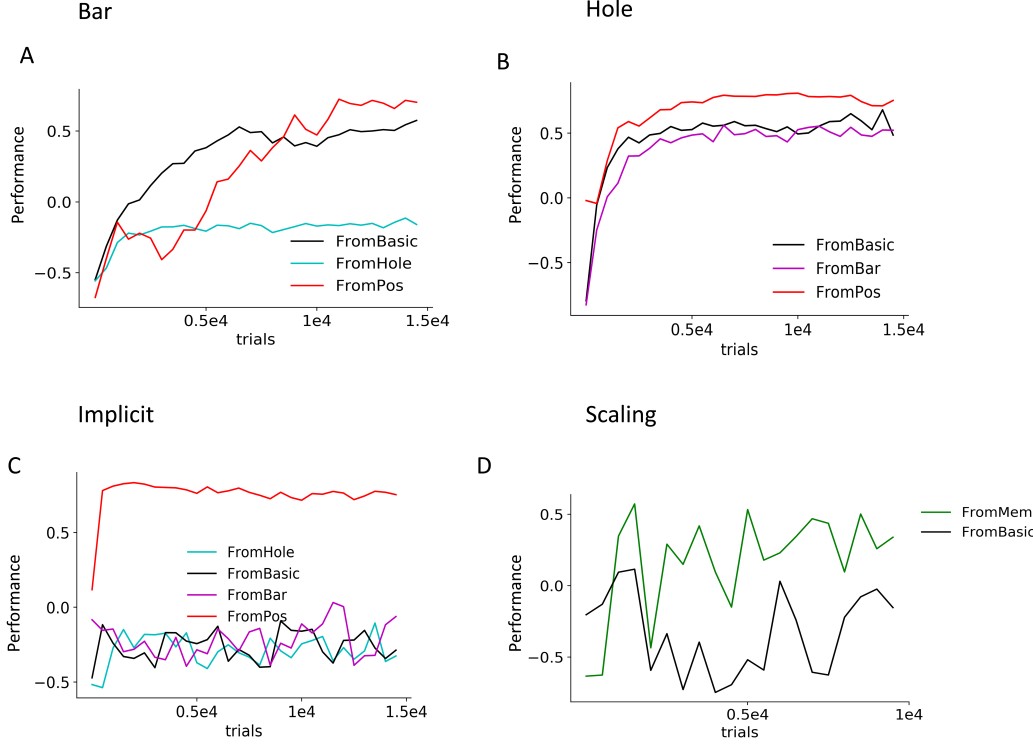

Figure 7: Generalization or transfer of end-to-end networks. Q-Learning of various tasks from a starting point of either pre-trained networks, or from end-to-end trained networks. The *FromPos* and *FromMem* curves denote PosNet and MemNet respecively. Each panel shows a learning curve for a different task.

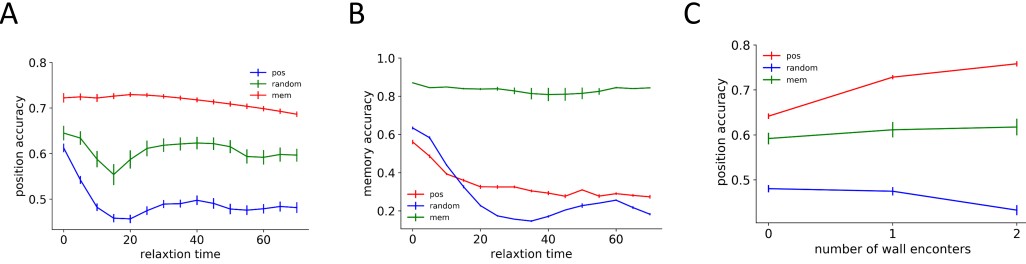

Figure 8: The accuracy of decoding position (**A**) or landmark memory (**B**) from the attractor manifold as a function of the number of relaxation steps. Panel A shows a drop in accuracy around $m = 25$, indicating that at this stage the process converges to irrelevant fixed points. (**C**) Conjunctive coding of memory and position. The accuracy of decoding position from the attractor manifold as a function of the number of wall encounters. Only PosNet shows an improvement in decoding with the added information. This is consistent with the joint representation of position and memory in the attractors.

$$S = -\alpha \sum_{i=1}^{n} \hat{P}(z_t) log P(z_t) - \beta \sum_{i=1}^{n} \hat{I}_t log P(I_t) - \gamma \sum_{i=1}^{n} \hat{A}_t log P(A_t) \qquad (17)$$

The agent was driven to explore an empty arena (with walls) using random actions, with a probability $p$ of changing action (direction) at any step. Table 1 shows the protocols (hyperparameters), Table

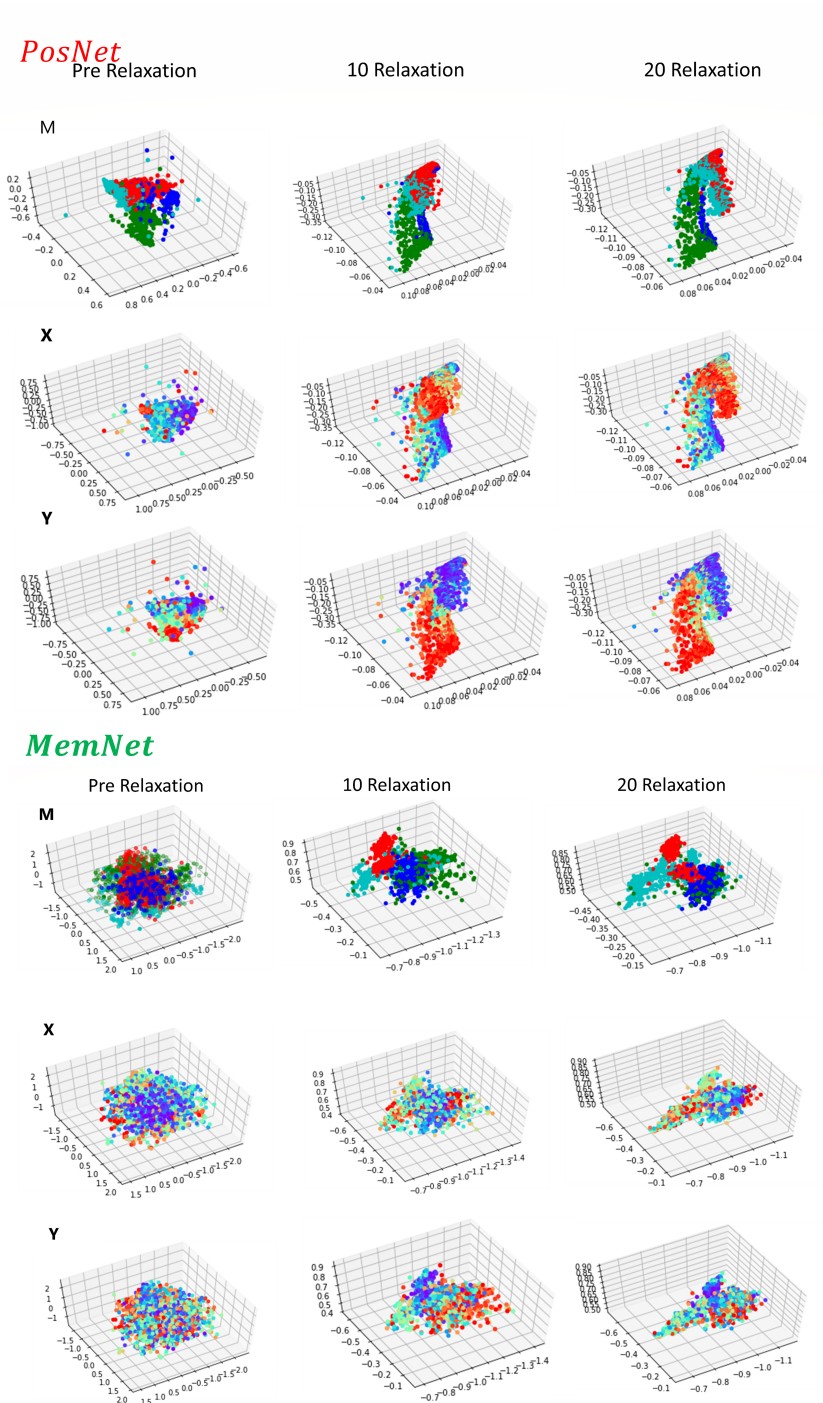

Figure 9: Slow points analysis for LSTM PosNet and MemNet(pretrained for landmark). Similar to Figure 4 in the main text. The three columns show points in phase space during a random walk and after 10 or 20 relaxation steps. The different rows are colored according to X, Y and identity of last wall. The PosNet is on top and MemNet is at bottom

2 shows the random networks hyperparameters, and table 3 shows the performance of the resulting networks on all tasks. For all pre-training protocols an l2 regularizer of $10^{-6}$ on internal weights

was used, and a learning rate of $10^{-5}$. All PosNet and MemNet training started from RandNet1 (detailed below).

Table 1: Pretraining protocols

| hyperparameters | | | |
|---|---|---|---|
| protocol name | loss($\alpha, \beta, \gamma$) | Weights adjusted | $p$ |
| PosNet1 | 1, 0, 0 | $W, W_a, W_i$ | 0.2 |
| PosNet2 | 1, 0, 0 | $W$ | 0.2 |
| PosNet3 | 1, 0, 0 | $W$ | 1 |
| PosNet4 | 1, 1, 0 | $W, W_a, W_i$ | 0.2 |
| PosNet5 | 1, 0.1, 0 | $W, W_a, W_i$ | 0.2 |
| MemNet1 | 0, 0.8, 0.2 | $W$ | 1 |
| MemNet2 | 0, 0.9, 0.1 | $W$ | 1 |
| MemNet3 | 0, 0.7, 0.3 | $W$ | 1 |
| MemNet4 | 0, 0.8, 0.2 | $W, W_a, W_i$ | 1 |
| MemNet5 | 0, 0.8, 0.2 | $W$ | 0.2 |

Different hyper parameter for RandNets:

$$h_{t+1} = (1 - \frac{1}{\tau})h_t + \frac{1}{\tau}(\tanh(Wh_t + W_i f(z_t) + W_a A_t + W_c C_t) \tag{18}$$

$$Q(h_t) = W_o h_t + b_o \tag{19}$$

Number of neurons used 512, time constant $\tau$ is taken to be 2, the choice of hyper parameters is according to standard reservoir computing litterature (Jaeger, 2010; Lukoševičius & Jaeger). The weights are taken from a standard Normal distribution. It is crucial to choose an appropriate standard deviation for success of training (Appendix section 5), which is summarized in 2, each unit represents $1/\sqrt{N}$

Table 2: Random Networks

| hyperparameters | | | |
|---|---|---|---|
| name | $W$ | $W_a$ | $W_i$ |
| RandNet1 | 1 | 1 | 10 |
| RandNet2 | 1 | 5 | 10 |
| RandNet3 | 0.5 | 1 | 10 |
| RandNet4 | 0.5 | 5 | 10 |
| RandNet5 | 1.2 | 1 | 10 |
| RandNet6 | 1.2 | 5 | 10 |

## 5.6 MODULAR NETWORK PROTOCOL

The results of modular network are obtained from combining the PosNet 1 and MemNet 25 from table . Both the learning of Q function and V function is learned in the the same way as main results equ (7,8,9, 10).

## 5.7 LINKING DYNAMICS TO CONNECTIVITY

Pretraining modified the internal connectivity of the networks. Here, we explore the link between connectivity and and dynamics. We draw inspiration from two observations in the field of reservoir computing (Lukoševičius & Jaeger, 2009). On the one hand, the norm of the internal connectivity has a large effect on network dynamics and performance, with an advantage to residing on the edge of chaos (Jaeger, 2010). On the other hand, restricting learning to the readout weights (which is then fed back to the network, Sussillo & Abbott (2009)) results in a low-rank perturbation to the connectivity, the possible contributions of which were recently explored (Mastrogiuseppe & Ostojic, 2018).

We thus analyzed both aspects. Fig. 10A shows the norms of several matrices as they evolve through pre-training, showing an opposite trend for PosNet and MemNet with respect to the internal

Table 3: Performance of all networks on all tasks

| protocol | BSC | HO | BAR | SC | SX | SY | IM |
|---|---|---|---|---|---|---|---|
| 1.PosNet1 | 0.91 | 0.79 | 0.73 | -0.36 | 0.90 | 0.19 | 0.78 |
| 2.PosNet1 | 0.97 | 0.78 | 0.79 | -0.47 | 0.89 | 0.53 | 0.11 |
| 3.PosNet1 | 0.95 | 0.74 | -0.04 | -0.16 | 0.70 | 0.70 | 0.66 |
| 4.PosNet1 | 0.97 | 0.82 | 0.68 | 0.03 | 0.89 | 0.21 | 0.69 |
| 5.PosNet1 | 0.88 | 0.74 | 0.73 | -0.36 | 0.88 | 0.20 | 0.78 |
| 6.PosNet1 | 0.91 | 0.63 | 0.48 | -0.41 | 0.62 | 0.04 | -0.21 |
| 7.PosNet1 | 0.94 | 0.64 | 0.51 | 0.26 | 0.64 | 0.15 | -0.31 |
| 8.PosNet2 | 0.89 | 0.62 | 0.14 | -0.25 | 0.76 | 0.24 | 0.84 |
| 9.PosNet3 | 0.93 | 0.48 | 0.41 | 0.04 | 0.22 | -0.24 | 0.73 |
| 10.PosNet4 | 0.83 | 0.58 | 0.72 | -0.07 | 0.16 | 0.58 | -0.22 |
| 11.PosNet5 | 0.86 | 0.69 | 0.48 | -0.19 | 0.21 | 0.65 | 0.01 |
| 12.PosNet6 | 0.80 | 0.17 | 0.11 | -0.33 | 0.07 | 0.36 | -0.10 |
| 13.RandNet1 | 0.89 | 0.52 | -0.15 | 0.26 | 0.79 | -0.17 | -0.14 |
| 14.RandNet2 | 0.84 | 0.25 | -0.89 | -0.03 | 0.62 | -0.29 | -0.10 |
| 15.RandNet3 | 0.51 | 0.05 | -0.93 | -0.39 | 0.30 | -0.35 | -0.09 |
| 16.RandNet4 | 0.65 | -0.31 | -0.94 | -0.37 | -0.23 | -0.04 | -0.17 |
| 17.RandNet5 | 0.91 | 0.22 | 0.15 | -0.16 | 0.62 | 0.06 | -0.47 |
| 18.RandNet6 | 0.88 | 0.70 | -0.36 | -0.44 | 0.63 | 0.11 | -0.33 |
| 19.MemNet1 | 0.65 | 0.27 | -0.84 | 0.62 | 0.92 | 0.62 | -0.20 |
| 20.MemNet1 | 0.79 | 0.15 | -0.94 | 0.48 | 0.64 | 0.43 | -0.14 |
| 21.MemNet1 | 0.82 | 0.28 | -0.30 | 0.37 | 0.69 | 0.45 | -0.15 |
| 22.MemNet1 | 0.73 | -0.09 | -0.51 | 0.65 | 0.84 | 0.58 | -0.29 |
| 23.MemNet1 | 0.84 | 0.54 | 0.39 | -0.07 | 0.85 | 0.26 | -0.41 |
| 24.MemNet2 | 0.76 | 0.52 | -0.87 | 0.58 | 0.47 | 0.90 | -0.29 |
| 25.MemNet3 | 0.76 | -0.11 | -0.46 | 0.65 | 0.83 | 0.61 | -0.28 |
| 26.MemNet4 | 0.73 | 0.35 | -0.64 | 0.43 | 0.91 | 0.50 | -0.10 |
| 27.MemNet5 | 0.75 | 0.08 | -0.86 | -0.37 | 0.12 | -0.12 | -0.04 |
| 28.End2End1 | 0.89 | 0.67 | 0.70 | -0.16 | -0.09 | 0.18 | 0.14 |
| 29.End2End2 | 0.67 | 0.8 | 0.73 | -0.62 | 0.51 | -0.61 | -0.36 |
| 30.Modular | 0.96 | 0.76 | 0.76 | 0.62 | 0.92 | 0.59 | 0.86 |

connectivity $W$. To estimate the low-rank component, we performed singular value decomposition on the *change* to the internal connectivity induced by pre-training (Fig. 10B).

$$W = W_0 + USV^T \tag{20}$$

The singular values of the actual change were compared to a shuffled version, revealing their low-rank structure (Fig. 10C,D). Note that pretraining was not constrained to generate such a low-rank perturbation. Furthermore, we show that the low-rank structure is partially correlated to the network's inputs, possibly contributing to their effective amplification through the dynamics (Fig. 10E-H). Because we detected both types of connectivity changes (norm and low-rank), we next sought to characterize their relative contributions to network dynamics, representation and behavior.

In order to assess the effect of matrix norms, we generated a large number of scaled random matrices and use the GLM analyse in figure 5 to access its influence on dynamics. We see the trade-off between landmark memory and path integration is affected by norm (Fig. 10E). But the actual numbers, however, are much lower for the scaled random matrices compared to the pretrained ones – indicating the importance of the low-rank component (Fig. 10F) . Indeed, when removing even only the leading 5 ranks from $\Delta W$, Network encoding and performance on all tasks approaches that of RandNet.

## 5.8 BEHAVIOR OF DIFFERENT NETWORKS

Different networks develop diverse strategies for metric and topological tasks. In this section, we give examples of typical trajectories of PosNet, MemNet, RanNet in the basic, bar and scaling tasks.

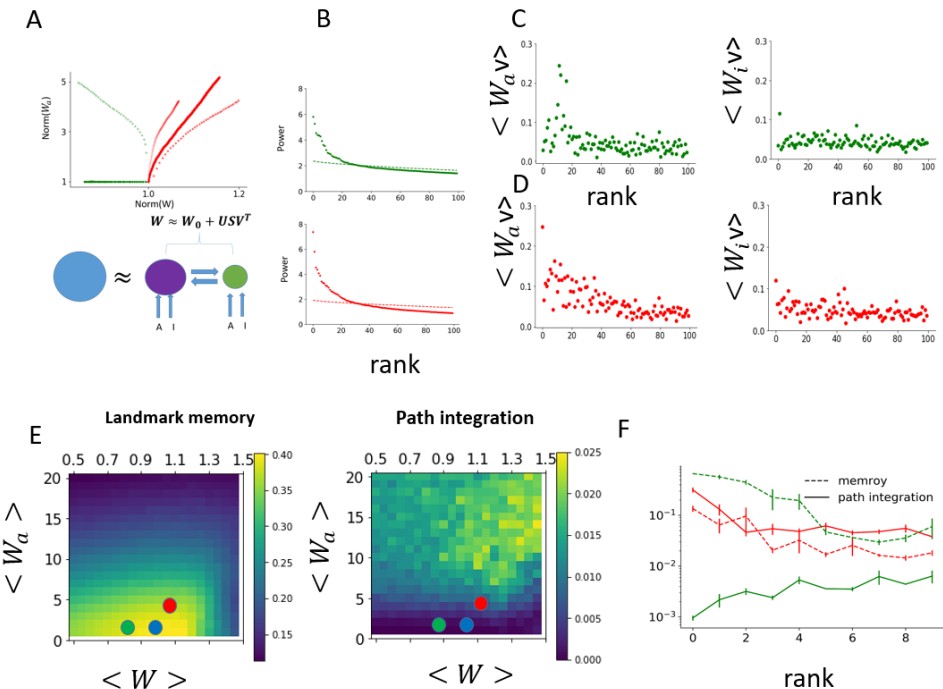

Figure 10: **Connectivity changes during pre-training**. **(A)** Top, Evolution of norm during pre-training for both PosNet (red) and MemNet (green). Bottom, low rank effect. The network can be decomposed into two parts, a random part, and a learned low-rank structure through SVD. **(B)** SVD of $\Delta W$ compared to a shuffled version of $\Delta W$, showing that most of the learned structure is concentrated in the first few ranks for PosNet(top).The same low-rank effect observed for MemNet(bottom). **(C-D)** Measuring the overlap between action feedback matrix or input matrix and output vector $v$ of $\Delta W$ for PosNet (C) and MemNet (D). **(E)** The variance of hidden states explained by a GLM model containing position and landmark memory. Each pixel represents a scaled random matrix, with the colored circles showing the norms of the pre-trained networks. The red, blue and green dots correspond to norm of selected PosNet, RandNet, MemNet for dynamics analysis. **(F)** Effect of gradually removing the leading ranks from $\Delta W$, measured by the variance explained in the GLM.

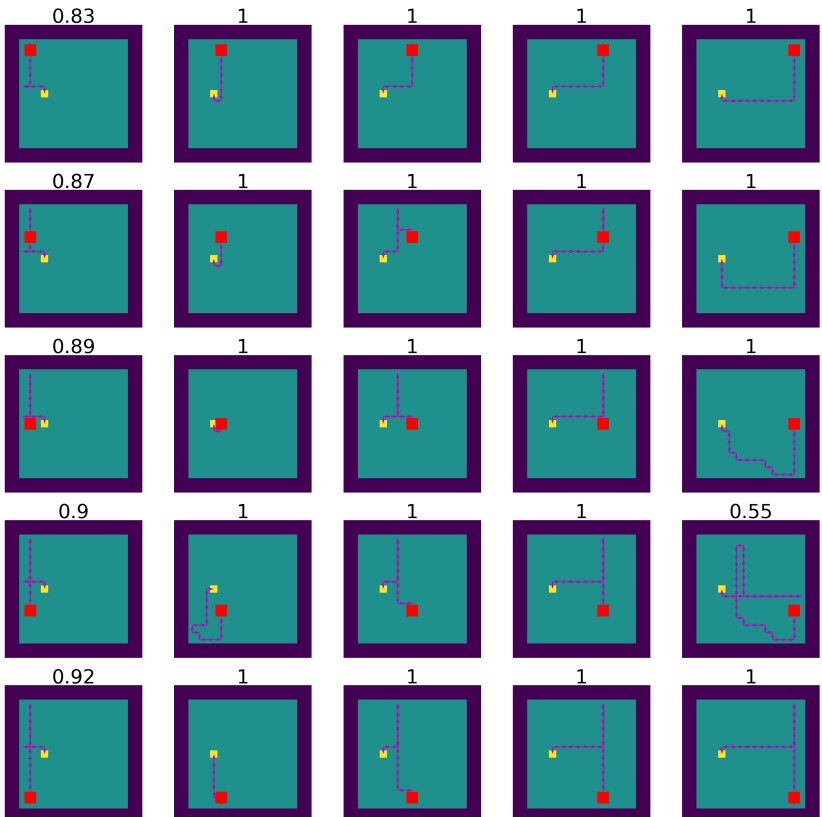

Figure 11: PosNet in basic task

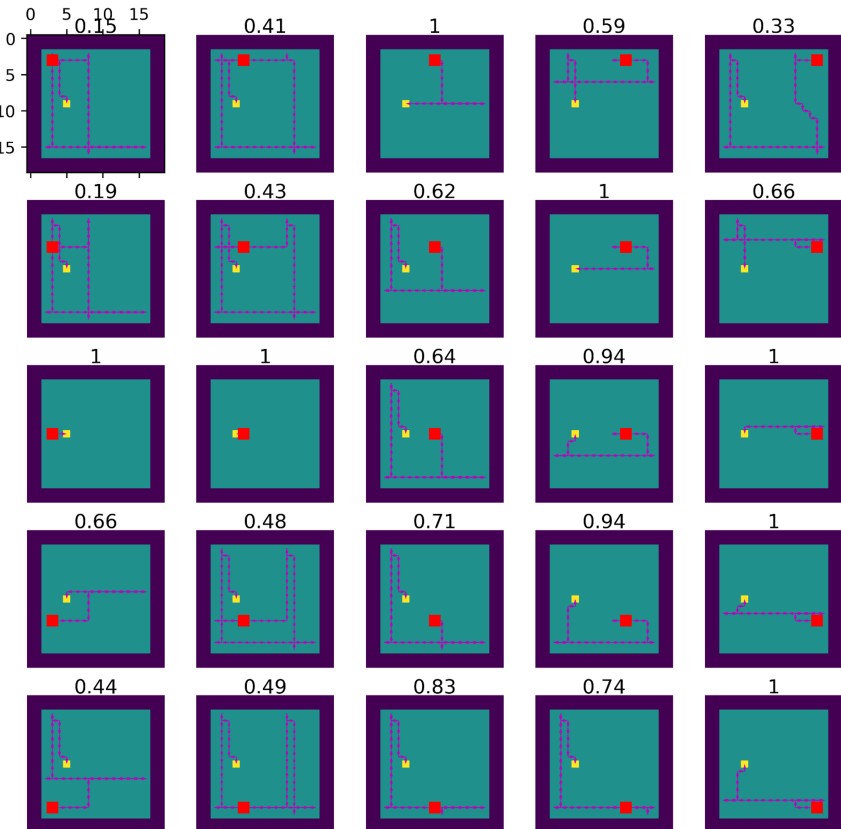

Figure 12: MemNet in basic task

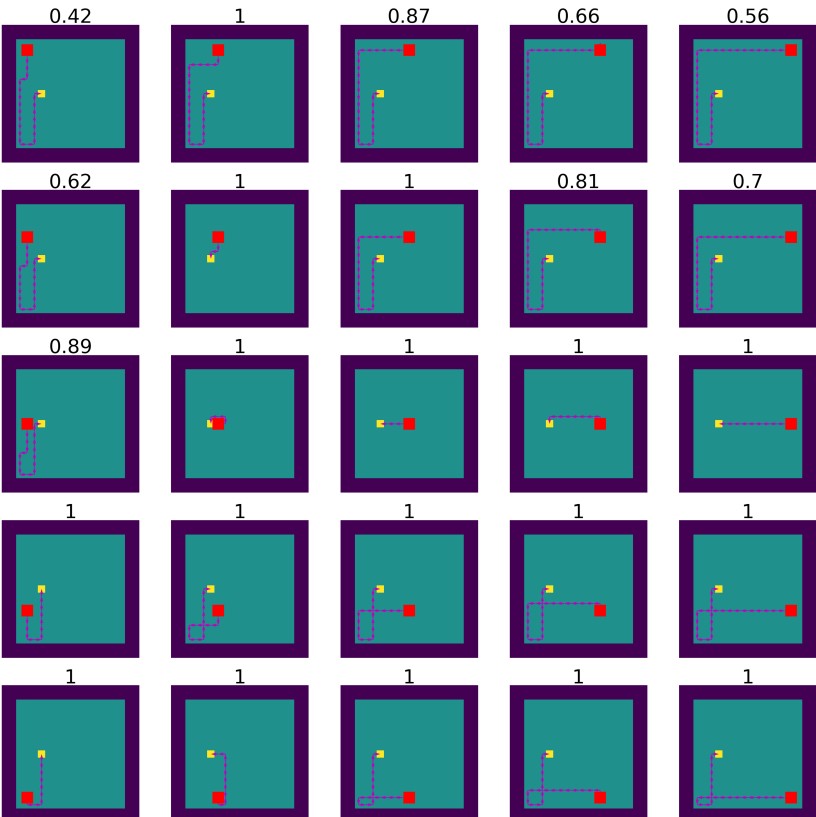

Figure 13: RanNet in basic task

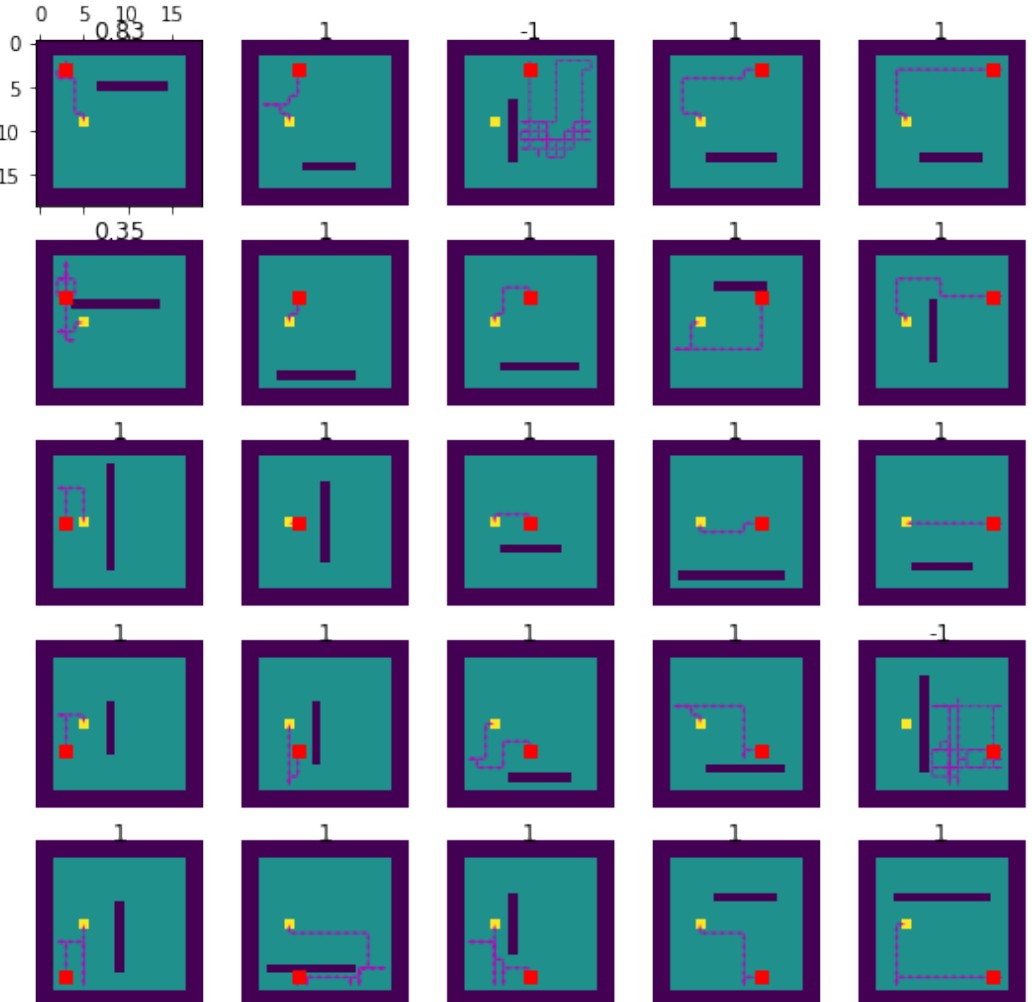

Figure 14: PosNet in bar task

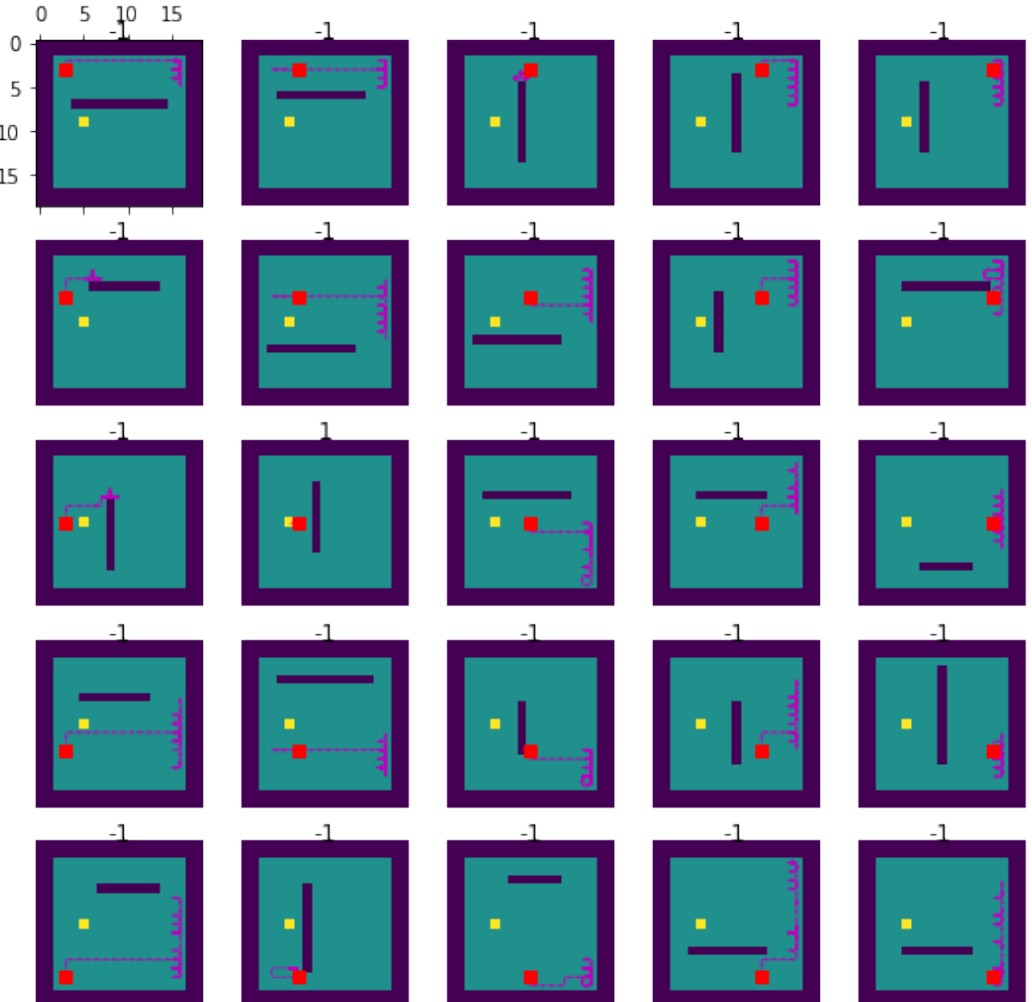

Figure 15: MemNet in bar task

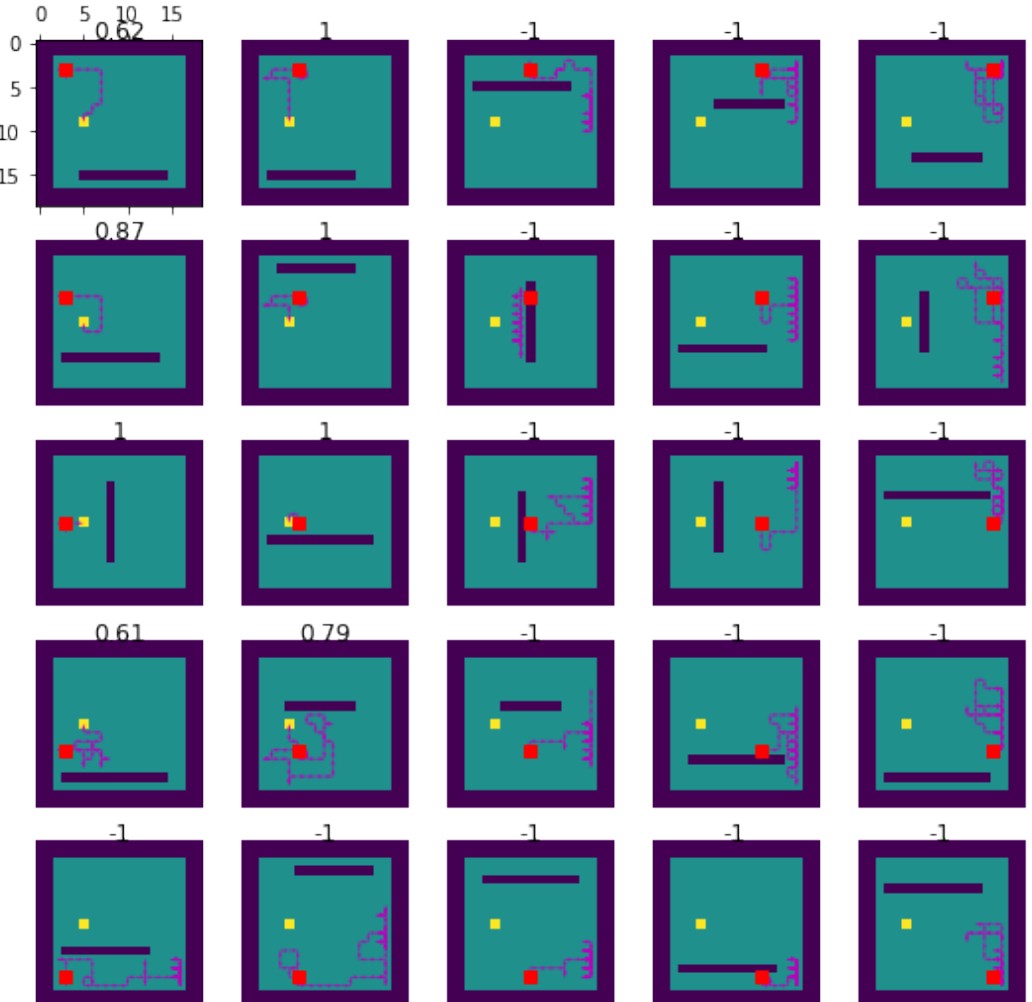

Figure 16: RanNet in bar task

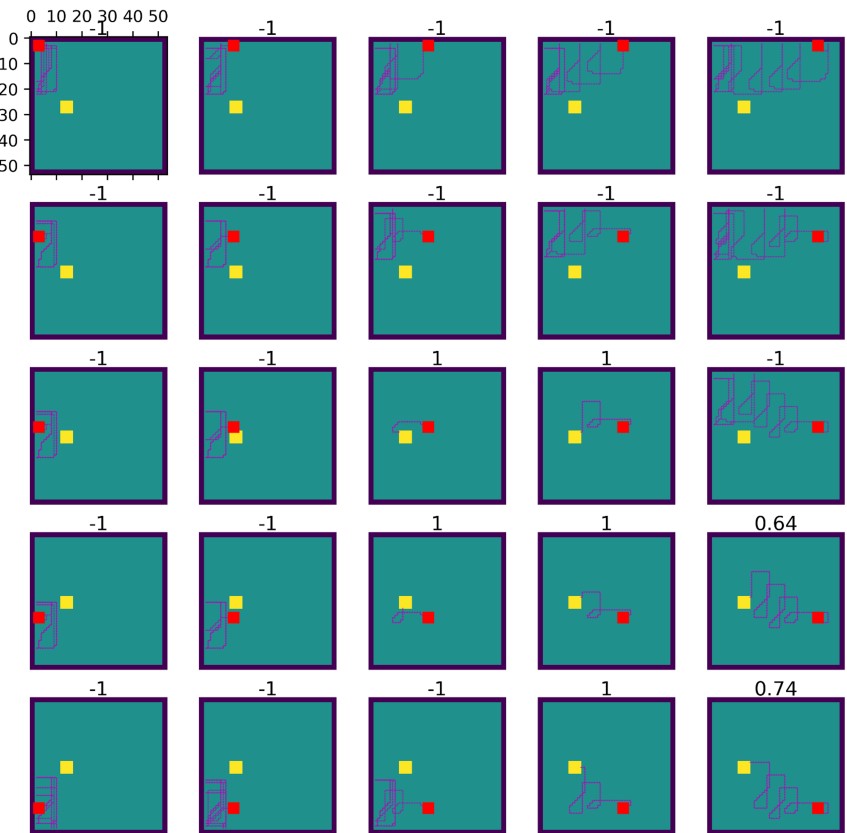

Figure 17: PosNet in scale task

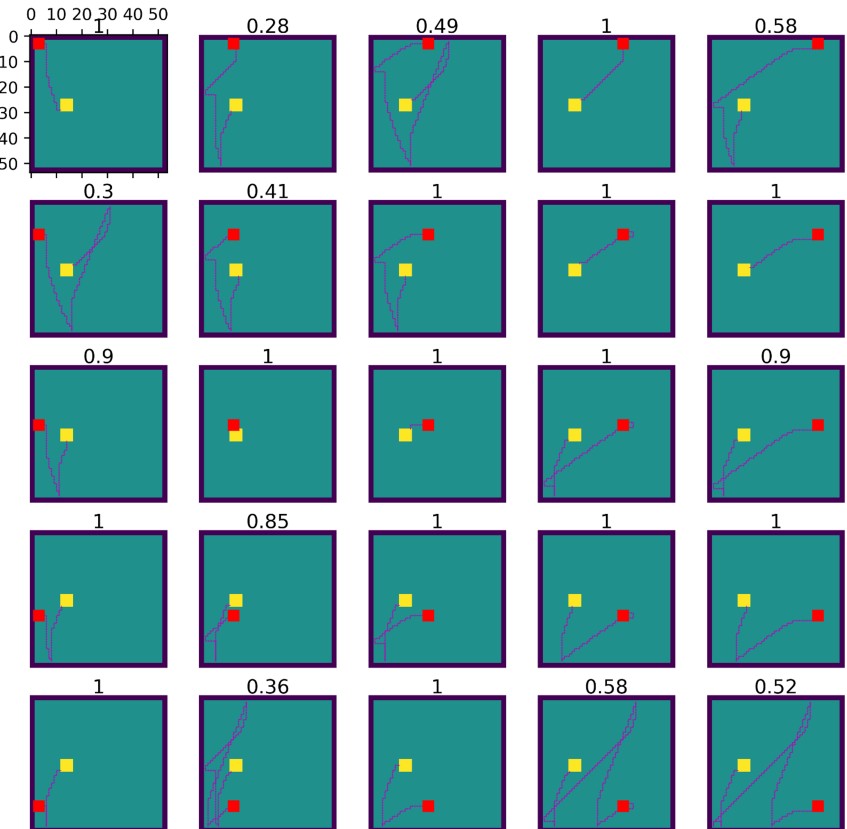

Figure 18: MemNet in scale task

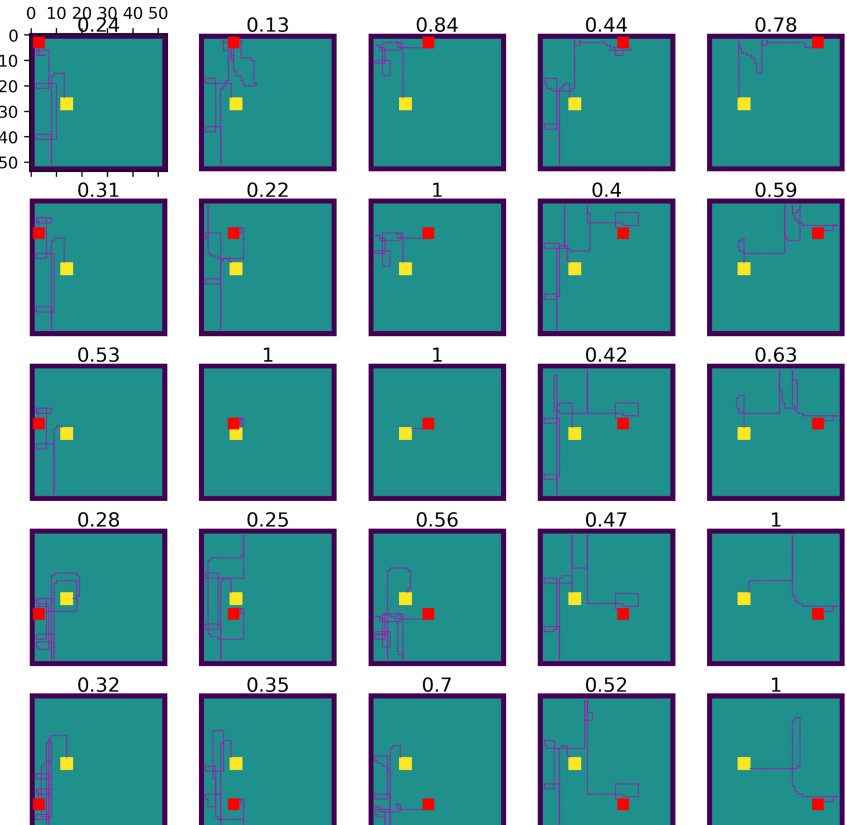

Figure 19: RanNet in scale task

