# OpenReview forum: "Implementing Inductive bias for different navigation tasks through diverse RNN attrractors"
_ICLR.cc/2020/Conference — Accept (Poster)_

### Official Review · AnonReviewer1 · 2019-10-23
**Official Blind Review #1**

**Rating:** 6

**Review:**

This paper studies the internal representations of recurrent neural networks trained on navigation tasks. By varying the weight of different terms in an objective used for supervised pre-training, RNNs are created that either use path integration or landmark memory for navigation. The paper shows that the pretraining method leads to differential performance when the readout layer of these networks networks is trained using Q-learning on different variants of a navigation task. The main result of the paper is obtained by finding the slow points of the dynamics of the trained RNNs. The paper finds that the RNNs pre-trained to use path integration contain 2D continuous attractors, allowing position memory. On the other hand, the RNNs pre-trained for landmark memory contain discrete attractors corresponding to the different landmarks.

An interesting implication of the findings for neuroscience is that the same underlying network architecture can learn different dynamics, explaining diverse types of navigation-related signals found in the mammalian brain (place cells, border cells etc.).

I am not entirely sure about the novelty or impact of the presented results. However, the exposition and the results are clear and it is interesting how pre-training can shape the dynamics of a network. I therefore recommend acceptance.

Minor comments:
- Please describe how the networks 1, 13, 20 used for Figure 4 were selected. Were they selected at random or selected according to some criteria?
- It may be interesting to study the effect of more modern RNNs, e.g. LSTMs or GRUs, on the dynamics.


**Experience Assessment:**

I do not know much about this area.

**Review Assessment: Checking Correctness Of Derivations And Theory:**

N/A

**Review Assessment: Checking Correctness Of Experiments:**

I assessed the sensibility of the experiments.

**Review Assessment: Thoroughness In Paper Reading:**

I read the paper at least twice and used my best judgement in assessing the paper.

---

> ### Author Response · Authors · 2019-11-08
> **Thanks for the feedback. Additional details provided.**
>
> We thank the reviewer for the interest in our work, and for the thoughtful comments.
>
> Q1) Please describe how the networks 1, 13, 20 used for Figure 4 were selected. Were they selected at random or selected according to some criteria?
> A1) The figures were chosen based on above-average performance, without a specific criterion. Visual inspection of other networks showed qualitatively similar results.
>
> Q2) It may be interesting to study the effect of more modern RNNs, e.g. LSTMs or GRUs, on the dynamics.
> A2) This will be done. See also answer to Q5 of reviewer #2, repeated here:
>
> R2.A5) We will train the gated architectures on the task and analyze them. Initially, we did use GRU for the task, and found that a vanilla RNN with the average effective timescale performs as well. We thus chose the “simpler” model. It is true that fixed point analysis can be equally applied to the more sophisticated architectures, and we will do this. Preliminary results show that pre-training LSTM networks for PosNet results in similar topology to the vanilla case.

---

> > ### Comment · AnonReviewer1 · 2019-11-14
> > **Response to authors.**
> >
> > Thanks for your comments.
> >
> > As pointed out by Reviewer 2, it is interesting that no networks seem to be good at both tasks. It may be worthwhile to expand the discussion of this phenomenon. Is this just a consequence of limited representational capacity of the network, or is there a more specific computational reason why accurate path integration and accurate landmark memory are mutually exclusive?

---

> > > ### Author Response · Authors · 2019-11-15
> > > **Speculation of reason for trade-off**
> > >
> > > Thanks for asking.  Following reviewer 2's suggestion, we added a modular system composed of PosNet, MemNet and an action selection module. This modular system was able to perform well on both topological and metric tasks, surpassing the performance of any individual network we trained.
> > >
> > > As for possible reasons for the tradeoff - the hidden states could represent the current position, stimulus memory, and other "irrelevant" features. Accurate path integration implies an ability to marginalize over stimulus memory and other features to obtain a clean version of the position. In principle, accurate decoding of the position could be done even if the hidden state contains all the other information. However, perhaps a more "natural" way is to allow the dynamics to suppress these aspects of the representation, and thus facilitate readout of the desired position information.
> > >
> > > Furthermore, the attractor landscape picture provides a dynamical reason for the tradeoff. Position requires a continuous attractor, whereas stimulus memory requires discrete attractors. It is possible to have four separated plane attractors, but perhaps easier to converge to one or the other.

---

### Official Review · AnonReviewer2 · 2019-10-24
**Official Blind Review #2**

**Rating:** 6

**Review:**

## Overview
This paper explores how pre-training a recurrent network on different navigational objectives confers different benefits when it comes to solving downstream tasks. First, networks are pretrained on an objective that either emphasizes position (path integration) or landmark memory (identity of the last wall encountered). This pretraining generates recurrent networks of two classes, called PosNets and MemNets (in addition to no pre-training, called RandNets). Surprisingly, the authors found that pre-training confers different benefits that manifests as differential performance of PosNets and MemNets across the suite. Some evidence is provided that this difference has to do with the requirements of the task. Moreover, the authors show how the different pretraining manifests as different dynamical structures (measured using fixed point analyses) present in the networks after pre-training. In particular, the PosNets contained a 2D plane attractor (used to readout position), whereas the MemNets contained clusters of fixed points (corresponding to the previously encountered landmark).

Overall, I thought this was a very interesting paper--it is one of the first papers I have seen that demonstrates how different pre-training requirements both change network dynamics (as measured by fixed points), and how those differences can yield different benefits on downstream navigational tasks.

## Major comments/concerns
- I think the presentation the pretraining objective (eq 3) could be clearer. Is eq 3 what is minimized during pre-training? How are \alpha, \beta, and \gamma chosen? \alpha is used to separate the two types of networks (MemNet from PosNet), which is the critical difference studied in the paper, so it would helpful to go into more detail about what \alpha controls and how it was chosen.

- For the first task, I am surprised that the agent is able to navigate the environment using only the eight neighboring locations. What is the size of the arena? What fraction of the states are simply surrounded on all sides by empty space? It would be informative to show some trajectories of agents solving the basic task.

- For Fig 3A and 3B, it would be nice to show the other network's performance (i.e. show the PosNet on the scaling task in 3A, and the MemNet on the bar task in 3B).

- How come there are no networks that are able to solve both sets of tasks? That is, how come there are no networks in the upper right region of Fig 3C? Does this suggest that an agent needs to combine two separate RNNs to solve the whole suite of tasks?

- What happens if you train recurrent networks with more sophisticated cell architectures (e.g. a GRU or an LSTM)? These are typically easier to train (and using automatic differentiation techniques are also amenable to fixed point analysis).

## Minor comments
- In eq. (1), use `\left(` and `\right)` to make the first set of parentheses have an appropriate height.
- Typo on the first line after eq. (6) (matrices)
- Relevant reference on comparing networks using dynamics around approximate fixed points: https://arxiv.org/abs/1907.08549.

**Experience Assessment:**

I have published one or two papers in this area.

**Review Assessment: Checking Correctness Of Derivations And Theory:**

I assessed the sensibility of the derivations and theory.

**Review Assessment: Checking Correctness Of Experiments:**

I carefully checked the experiments.

**Review Assessment: Thoroughness In Paper Reading:**

I read the paper thoroughly.

---

> ### Author Response · Authors · 2019-11-08
> **Thanks for the feedback. Additional details provided.**
>
> We thank the reviewer for appreciating the novelty of our work, and for the thoughtful comments.
>
> Q1) I think the presentation the pretraining objective (eq 3) could be clearer. Is eq 3 what is minimized during pre-training? How are \alpha, \beta, and \gamma chosen? \alpha is used to separate the two types of networks (MemNet from PosNet), which is the critical difference studied in the paper, so it would helpful to go into more detail about what \alpha controls and how it was chosen.
>
> A1) This part will be rewritten for clarity. Yes - Eq. 3 is minimized during pre-training. The ratio between alpha and beta controls the relative strength/tradeoff of memory and position. We found that, in general, with $\alpha$ larger than 0 , the position representation dominates over the memory one. The exact values were chosen manually through trial and error.
>
>
> Q2) For the first task, I am surprised that the agent is able to navigate the environment using only the eight neighboring locations. What is the size of the arena? What fraction of the states are simply surrounded on all sides by empty space? It would be informative to show some trajectories of agents solving the basic task.
> A2) We were initially also surprised by this. The size is 15 (will be written clearly), which means that 161 (8 reward + 56 border) out of 225 are surrounded by empty space (71.5 %).  The agent is able to perform the task because it collects information into its recurrent dynamics. The information is gathered continuously in the trajectory, and not only from wall encounters. For instance, in many trials the agent reaches the reward before touching the wall twice – that is, without full metric information.  Regarding the size of arena,   we were able to train networks on the topological tasks for large arenas (50x50, 92% empty). We will include several example trajectories in the supplementary material, illustrating the various ways in which the agent solves the task.
>
> Q3) For Fig 3A and 3B, it would be nice to show the other network's performance (i.e. show the PosNet on the scaling task in 3A, and the MemNet on the bar task in 3B).
> A4) We will add this.
>
> Q4) How come there are no networks that are able to solve both sets of tasks? That is, how come there are no networks in the upper right region of Fig 3C? Does this suggest that an agent needs to combine two separate RNNs to solve the whole suite of tasks?
> A4) We think this is indeed the case (as we mention in the discussion). We tried to carefully select networks (post-hoc) and could get a partial improvement towards the upper right, but still there was a tradeoff.
> We will also train a modular network to perform the task, but are not sure whether the results will be ready in one week.
>
> Q5) What happens if you train recurrent networks with more sophisticated cell architectures (e.g. a GRU or an LSTM)? These are typically easier to train (and using automatic differentiation techniques are also amenable to fixed point analysis).
> A5) We will train the gated architectures on the task and analyze them. Initially, we did use GRU for the task, and found that a vanilla RNN with the average effective timescale performs as well. We thus chose the “simpler” model. It is true that fixed point analysis can be equally applied to the more sophisticated architectures, and we will do this. Preliminary results show that pre-training LSTM networks for PosNet results in similar topology to the vanilla case.
>
> Q6) ## Minor comments
> - In eq. (1), use `\left(` and `\right)` to make the first set of parentheses have an appropriate height.
> - Typo on the first line after eq. (6) (matrices)
> - Relevant reference on comparing networks using dynamics around approximate fixed points:
> A6) Will be fixed.

---

### Official Review · AnonReviewer3 · 2019-10-25
**Official Blind Review #3**

**Rating:** 3

**Review:**

This paper presents a method for the navigation task. The proposed method is inspired by the concept of cognitive map in human and animal.
A recurrent neural network is incorporated and training is divided in two steps of (1) task-agnostic pre-training and (2) task-speciﬁc Q learning.

The paper is well-written and clear.

While the idea of using a representation inspired by cognitive maps is interesting, the paper does not offer much technical novelty. (e.g no technical novelty in eq. 1 and eq. 2)

The experimental results are weak and only a simple domain is tested.

It is not clear how efficient the method would be compared to other approaches.

Visualizations can be improved. As an example, Fig. 4 is not quite self-representative.

I see the paper has a large room for improvement and the current manuscript is not convincing for publication.


**Experience Assessment:**

I have published in this field for several years.

**Review Assessment: Checking Correctness Of Derivations And Theory:**

I assessed the sensibility of the derivations and theory.

**Review Assessment: Checking Correctness Of Experiments:**

I assessed the sensibility of the experiments.

**Review Assessment: Thoroughness In Paper Reading:**

I read the paper at least twice and used my best judgement in assessing the paper.

---

> ### Author Response · Authors · 2019-11-08
> **Clarification of novelty**
>
>
> We thank the reviewer for the thoughtful comments on the paper.
> Below, we explain what we view as the novelty in the paper and describe some new results that we hope strengthen the argument.
>
> Q1) While the idea of using a representation inspired by cognitive maps is interesting, the paper does not offer much technical novelty. (e.g no technical novelty in eq. 1 and eq. 2)
>
> A1) These equations are indeed not novel – they are the dynamics of a simple RNN and appear in the “Task Definition” section where we define the framework. The novel aspects, which appear later, are: (1) Using a neuroscience inspired pre-training protocol. (2) Showing how pretraining biases subsequent RL performance on different tasks. (3) Uncovering the mechanistic reason for this bias. Namely, different arrangements of slow points in the phase space of the recurrent network.
>
> Q2) The experimental results are weak and only a simple domain is tested.
> A2) We chose a simple domain to increase our analysis power. In particular, this choice enabled a more systematic study of many tasks and networks than would be feasible for a more complex setting. Our choice of a simple domain also allowed for an easier interpretation of the slow point analysis which revealed that the underlying mechanism behind the effect of pre-training is the dynamical objects. The correlation between slow points and task features is easier to see in our setting.
> Pretraining is also used in other domains such as NLP or image recognition (Devlin et al. 2018, You et al. 2015). We believe that obtaining deeper understanding in a simple setting could pave the way for implementations in other domains in the future.
>
>
> As for the weakness of the experimental results – we are not sure we completely understand this comment and would welcome clarification. Figures 2 and 3 clearly show a tradeoff between the two pretraining protocols, and figure 4 shows different dynamical objects (the clarity of this figure will be improved). These are the two main results, and we believe the data support them.
>
> Q3) It is not clear how efficient the method would be compared to other approaches.
> A3) We performed additional experiments that address this point. We trained the network using end-to-end learning, without pretraining. This was done both by extending our Q-learning to start from random networks, and by an adaptation of a different method (DDPG, Heess et al. ). These experiments show that: (1) Training in our pre-training protocol is faster. (2) Performance in our pre-training protocol is much better for the topological tasks (end-to-end was not able to learn it), and it is comparable in most metric tasks.
> We believe these new results strengthen the case for the importance of inductive bias in the form of discrete fixed points to learn the topological tasks.
>
> Q4) Visualizations can be improved. As an example, Fig. 4 is not quite self-representative.
> A4) We thank the reviewer for this comment. The visualizations will be improved. In particular, we will better illustrate the main points of figure 4.
>
> References:
> Heess, N., Hunt, J. J., Lillicrap, T. P., & Silver, D. (2015). Memory-based control with recurrent neural networks. arXiv preprint arXiv:1512.04455.
>
> Devlin, J., Chang, M. W., Lee, K., & Toutanova, K. (2018). Bert: Pre-training of deep bidirectional transformers for language understanding. arXiv preprint arXiv:1810.04805.
>
> You, Q., Luo, J., Jin, H., & Yang, J. (2015, February). Robust image sentiment analysis using progressively trained and domain transferred deep networks. In Twenty-ninth AAAI conference on artificial intelligence.

---

### Author Response · Authors · 2019-11-15
**Revised version uploaded: New results, improved clarity.**

We thank all reviewers for the discussion and believe that our paper has improved through this process.
We now uploaded a revised version of the paper, that we hope answers all the concerns raised by the reviewers. In particular both points #1 and #2 address the novelty concerns raised by reviewer #3.

The main changes are:

1) Following reviewer 2's suggestion, we added a modular system composed of PosNet, MemNet and an action selection module. This modular system was able to perform well on both topological and metric tasks, surpassing the performance of any individual network we trained.  The success of modular system roots in understanding of tradeoff postulated in most part of paper.

2) We compared our two-stage learning to a direct end-to-end protocol, in which we modify the entire connectivity to perform the various tasks. The main conclusions from this comparison are: (A) Two-stage learning is faster than end-to-end. (B) Good performance on topological tasks could only be achieved with two-stage learning.(C) Performance on metric tasks(except implicit context) is similar.  (D) Generalization (transfer) to new tasks is better for two-stage learning. This implies potential to use our methods in other multiple tasks learning(meta learning) problems.

3) We repeated our dynamics analysis for LSTM units, revealing qualitatively similar behavior.

4) We improved the clarity of text and figures:
4.1) Example trajectories are provided in figure 2 and in the appendix.
4.2) Figure 4 was remade, and the slow point concept illustrated with a cartoon.
4.3) Pre-training description was improved.

5) We added in the discussion part: possible reason for trade-off between different dynamics according to request of both reviewer 1 and 2.

6) Note that the new experiments (end-to-end, modular, LSTM) were only done for a small number of networks due to the time constraint, and we will gather more statistics for the final version.

We hope these changes strengthen the paper in the eyes of the reviewers.
We will be happy to answer any further questions.

Sincerely,
The authors.

---

### Decision · Program_Chairs · 2019-12-19

**Decision:**

Accept (Poster)

**Comment:**

Navigation is learned in a two-stage process, where the (recurrent) network is first pre-trained in a task-agnostic stage and then fine-tuned using Q-learning. The analysis of the learned network confirms that what has been learned in the task-agnostic pre-training stage takes the form of attractors.

The reviewers generally liked this work, but complained about lack of comparison studies / baselines. The authors then carried out such studies and did a major update of the paper.

Given that the extensive update of the paper seems to have addressed the reviewers' complaints, I think this paper can be accepted.